# An International Non-Inferiority Study for the Benchmarking of AI for Routine Radiology Cases: Chest X-ray, Fluorography and Mammography

**DOI:** 10.3390/healthcare11121684

**Published:** 2023-06-08

**Authors:** Kirill Arzamasov, Yuriy Vasilev, Anton Vladzymyrskyy, Olga Omelyanskaya, Igor Shulkin, Darya Kozikhina, Inna Goncharova, Pavel Gelezhe, Yury Kirpichev, Tatiana Bobrovskaya, Anna Andreychenko

**Affiliations:** 1State Budget-Funded Health Care Institution of the City of Moscow “Research and Practical Clinical Center for Diagnostics and Telemedicine Technologies of the Moscow Health Care Department”, Petrovka Street, 24, Building 1, 127051 Moscow, Russiabobrovskayatm@zdrav.mos.ru (T.B.);; 2Federal State Budgetary Institution “National Medical and Surgical Center Named after N.I. Pirogov” of the Ministry of Health of the Russian Federation, Nizhnyaya Pervomayskaya Street, 70, 105203 Moscow, Russia; 3Department of Information and Internet Technologies, I.M. Sechenov First Moscow State Medical University of the Ministry of Health of the Russian Federation (Sechenov University), Trubetskaya Street, 8, Building 2, 119991 Moscow, Russia

**Keywords:** stand-alone artificial intelligence, radiology, benchmarking, population screening

## Abstract

An international reader study was conducted to gauge an average diagnostic accuracy of radiologists interpreting chest X-ray images, including those from fluorography and mammography, and establish requirements for stand-alone radiological artificial intelligence (AI) models. The retrospective studies in the datasets were labelled as containing or not containing target pathological findings based on a consensus of two experienced radiologists, and the results of a laboratory test and follow-up examination, where applicable. A total of 204 radiologists from 11 countries with various experience performed an assessment of the dataset with a 5-point Likert scale via a web platform. Eight commercial radiological AI models analyzed the same dataset. The AI AUROC was 0.87 (95% CI:0.83–0.9) versus 0.96 (95% CI 0.94–0.97) for radiologists. The sensitivity and specificity of AI versus radiologists were 0.71 (95% CI 0.64–0.78) versus 0.91 (95% CI 0.86–0.95) and 0.93 (95% CI 0.89–0.96) versus 0.9 (95% CI 0.85–0.94) for AI. The overall diagnostic accuracy of radiologists was superior to AI for chest X-ray and mammography. However, the accuracy of AI was noninferior to the least experienced radiologists for mammography and fluorography, and to all radiologists for chest X-ray. Therefore, an AI-based first reading could be recommended to reduce the workload burden of radiologists for the most common radiological studies such as chest X-ray and mammography.

## 1. Introduction

A steadily increasing volume of prescribed radiological diagnostic examinations and the increasing amount of diagnostic equipment has skyrocketed the workload of radiologists [1]. More than half of all radiology studies are composed of mammography, chest X-ray and chest fluorography [2,3]. The WHO (World Health Organization) guidelines evaluated the reduction in mortality due to mammography screening and consider it to be the most widely used and valuable noninvasive method for early breast cancer detection [4,5]. Chest X-ray (and fluorography in certain countries) is widely used for routine, emergency and screening purposes. During the pandemic, it became even more valuable, as the WHO recommends including chest X-ray into a diagnostic approach for patients suspected of the 2019 novel coronavirus disease (COVID-19) [6,7].

Due to the rapidly increasing number of radiological examinations, the application of artificial Intelligence (AI) models for carrying out a first reading becomes valuable in order to reduce radiologists’ workload and improve diagnostic accuracy in the absence of experienced specialists [8,9]. Recent studies have demonstrated that the diagnostic accuracy of AI models for medical imaging approaches the performance of medical experts and even outperforms them in several fields [10]. This has led to a rapid explosion of commercially available and registered AI models for mammography and thoracic radiology (including chest X-ray and fluorography) analysis [11,12]. Thus, AI models are actively integrated into the radiology workflow. However, while the accuracy metrics claimed by the developers are quite high, their real-world performance must be carefully evaluated and compared with the radiologist’s performance in order to ensure practical value and safety [3,4,13,14]. Therefore, the WHO warns that, despite obvious benefits, the deployment of AI in medicine is fraught with risks that should be minimized [15].

At the same time, there are controversial opinions on the use of AI models compared to radiologists with various levels of experience. However, studies have demonstrated the ability of AI models to reach the levels of radiologists’ performance [13]. On the other hand, the accuracy of mammography interpretation by experienced radiologists varies highly [16] and previous-generation CAD systems have not significantly improved the accuracy of mammography readings [17]. Only several studies [1,7,18] have assessed the accuracy of AI algorithms in relation to the analysis of chest X-rays; there are almost no data for fluorography. The expectations of AI implementation for X-ray imaging thus remain ambiguous. Even promising results of AI models with high accuracy metrics are associated with limited specificity for the classification of the particular findings. Thus, it can barely be a substitute for a radiologist [7,19,20].

Since variations in radiologists’ performance are widely observed, they can lead to different results in the comparison between radiologists and AI. This may compromise the objective assessment of a particular AI model, or lead to a misjudgment of the benefits and limitations of AI for the whole field of medical imaging. Therefore, multicenter and international studies with different groups of radiologists are of particular interest for the benchmarking of AI and human readers. A study carried out among 101 radiologists from 7 countries demonstrated that the precision of the AI model accuracy for identifying breast cancer with mammography was comparable to that of an average radiologist [13]. Another major study with more than 1100 participants from 44 countries revealed that none of the AI models could outperform radiologists, but the combination of a single reader evaluation with AI results improved the total accuracy of screening mammography [4]. A recent international study demonstrated that an AI model could exceed the average performance of mammography specialists, but the comparison was performed for a relatively small group of six readers [14]. Undoubtedly, larger international studies are needed in order to establish an unbiased comparison of AI and human readers’ performance. This study aimed to determine the average diagnostic accuracy of radiologists interpreting chest X-ray images, including those from fluorography and mammography, on an international scale for benchmarking with the stand-alone AI model’s performance metrics for the same cases.

## 2. Materials and Methods

The retrospective study was conducted according to Standards for Reporting Diagnostic Accuracy Studies (STARD) 2015 guidelines. The overall study design scheme is shown in Figure 1.

### 2.1. Reference Dataset

The reference dataset (local test dataset) was collected retrospectively from the radiological exams performed in outpatient Moscow state medical facilities for screening and diagnostic purposes in 2018–2019. The dataset contained studies marked as ‘without (target) pathology’ and ‘with (target) pathology’. The target pathology was defined based on a list of pathological radiological findings compiled based on their clinical significance and frequency of occurrence in the routine practice of radiologists. All studies were selected based on electronic medical records and then double-checked by radiology experts who had at least 5 years of experience in thoracic radiology or breast imaging. Pathomorphological confirmation for malignancies was derived from electronic medical records. Table 1 contains details on the dataset.

The following target pathologies (terminology proposed by the Fleischner society [21]) for digital chest radiography and fluorography were included in this study:Pneumothorax;Atelectasis;Nodules or mass;Infiltrate or consolidation;Miliary pattern, or dissemination;Cavity;Pulmonary calcification;Pleural effusion;Fracture, or rupture of the bone cortical layer.

For digital mammography, a target pathology was defined based on the corresponding malignancy probability classifications of BI-RADS3-5 on the diagnostic scale or BI-RADS0 on the screening scale [22], with the confirmed diagnosis based on biopsy results or a follow-up negative MMG study for BI-RADS1-2. The right and the left breasts were assessed separately; however, the study was marked as pathological if the signs of the pathology were detected in at least one breast.

Inclusion criteria for the study: (1) all studies in the dataset were presented in Digital Imaging and Communications in Medicine (DICOM) format and anonymized; (2) sufficient number and appropriate diagnostic quality of the images was required for every study: a chest X-ray and a digital fluorography study included an anterior–posterior view; mammography studies contained the breasts images in two views (craniocaudal and mediolateral); and (3) for the target pathological findings, the truthing included: (a) histological confirmation of the malignancy presence and a follow-up study without the pathological findings for the absence of the malignancy or (b) a double consensus between two expert radiologists for all other findings.

Exclusion criteria included: (1) lung or breast surgery; (2) additional opacifications from medical devices, clothing or extracorporeal objects; (3) technical defects of the image and/or the positioning; (4) absence of histological or expert confirmation of the pathology; and (5) age < 18 years.

### 2.2. AI Models

The study included eight AI models that participated in the experiment with the use of innovative computer vision technologies for medical image analysis and subsequent applicability in the healthcare system of Moscow (https://mosmed.ai/en/, accessed on 1 June 2023). This research was registered in ClinicalTrials (NCT04489992). The study included commercial AI models to identify pathological signs on digital chest radiography (4 AI models [23,24,25,26]), fluorography (2 AI models [24,27]) and mammography (2 AI models [28,29]). The criterion for inclusion of these models was full compliance with the use cases, i.e., each AI provider declared detection of all radiological findings included in the use case: a list of the lung pathologies for chest X-ray and fluorography, and breast cancer signs corresponding to BI-RADS0 for mammography. As was reported by the AI models’ developers, diagnostic accuracy metrics corresponded to those of current state-of-art AI for the use cases [7,13,30,31,32,33,34,35]. All of the AI models provided responses per study as a general abnormality score (range 0–1) without providing details on the findings. Consequently, this did not allow us to assess the performance per finding. The AI models were deployed as stand-alone systems. The details of AI models are provided in the Appendix A.

In the present study, we did not conduct any refinement of the AI models. Instead, we exclusively utilized off-the-shelf commercial solutions as they were provided by the developers. It is important to note that no modifications or alterations were made to the AI models during the course of this study.

### 2.3. Web Platform for Conducting the Reader Study

For the human reader study, we developed a web-based platform in order to let participants evaluate cases online. A participant could determine a start date for every use case; however, the duration of interpretation was fixed to 3 days from the start date. The participants also chose the number of studies for interpretation—20, 50 or 80 studies. In order to ensure the representativeness of the real practice performance of radiologists, they started the reader study after completing training using the web-based platform for five cases that were not included in the final evaluation.

This study aimed to determine the average accuracy of radiologists to benchmark AI-performed radiological evaluation as a stand-alone service. Therefore, in this study, we compared the diagnostic accuracy metrics of a radiologist without AI, and AI on its own as a first or second independent reading. To create equal conditions for AI models and radiologists, there was no additional information provided, such as complaints or medical history (Figure 2). Patients’ age and sex were available to radiologists, but no radiological report or clinical information was provided. Radiologists did not write a detailed clinical report. They only identified findings and rated their confidence in the presence of each case for the presence of any pathological findings using a five-point scale (from 1—definitely without pathology to 5—definitely with pathology) similar to that used in other reader studies [35]. Age and sex were also provided to the AI models as DICOM tags. Whether these data were used by AI models is not known. Providing similar data to radiologists and AI models ensured an objective comparison of their performance in the clinical scenario when AI performed an independent reading.

In the upper part of the control panel of the Web platform, there were always two buttons that opened dialog boxes containing: (1) the platform user guide and (2) diagnostic criteria, according to which the participant should classify a study as normal (‘without pathology‘) or abnormal (‘with pathology‘)—these corresponded to the AI-based triage. The following options were given for scoring a study on the panel by a human reader:Definitely without pathology (probability of pathology = 0.0);Probably without pathology (probability of pathology = 0.25);Undefined (questionable/unreadable) (probability of pathology = 0.5)Probably with pathology (probability of pathology = 0.75);Definitely with pathology (probability of pathology = 1.0);

### 2.4. Participating Radiologists

A total of 204 radiologists from 11 countries participated in the study. Some of these participants (n = 96) specialized in breast imaging. Table 1 presents the distribution of radiologists by the use cases with an indication of their experience and country. Each radiologist had access to the reader study datasets of three modalities—chest X-ray, chest fluorography, mammography. The evaluation of studies was conducted by every radiologist independently from 27 November to 13 December 2020. Every radiologist was provided with a set of 20/50/80 cases of the same modality (i.e., chest X-ray, fluorography or mammography) according to his/her choice. Each case and set of cases could be evaluated only once to ensure the uniqueness of responses. A user could complete the interpretation for several modalities. The exclusion criteria for radiologists were as follows: (1) the registration form was not completed (no information regarding experience, employment, preferred modality); and (2) absence of responses.

### 2.5. Score Analysis: Determination of the Consensus Score for Radiologists and AI Models

Studies in which most scores (>50%) were “undefined” were excluded from further analysis [36]. For the remaining studies, a consensus score was defined based on the median score of the readers. In the case of a tie of frequencies, the higher score was selected. If the case had less than 5 responses it was also excluded from further evaluation with an exception for breast imaging specialists. A consensus score between AI models for each case was reached in the following way: First, the responses for each AI model were calibrated individually in order to combine the probability values of each AI model. Second, an average probability score of all AI models was set as a consensus score for each case.

### 2.6. Statistical Analysis

The performance of AI and the radiologists was assessed by generating a receiver operating characteristic (ROC) curve. The area under the ROC curve (AUROC) was reported with 95% confidence intervals (95% CI). The Delong method was used to calculate the confidence interval for the AUROC [37]. A smoothing was used to build the ROC curve.

To conduct the ROC analysis, we required a binary estimation (true value) as well as the output from the “classifier”. When evaluating AI algorithms, we utilized the pathology probability value as the input, which ranged from 0 to 1, with a precision of 0.01. Similarly, when evaluating a radiologist’s performance, we also employed the probability values assigned by the radiologist. However, it is challenging for a radiologist to precisely assign a digital probability value for the presence of pathology. Therefore, we employed a more comprehensible gradient scale that could be easily converted into absolute values: “definitely without pathology” = 0, and “definitely with pathology” = 1. Subsequently, a standard ROC analysis was conducted to determine the diagnostic accuracy indicators.

A *p*-value for the AUROC was calculated using a permutation test [38]. A *p*-value less than 0.05 was considered to represent a significant difference. The null hypothesis was that the AUROC of AI and an average human reader were the same. The analysis did not account for the variability between the individual radiologists. A maximum of the Youden Index was used to determine an optimal cut-off value for the radiologists and AI metrics [39,40]. ROC analysis was used to select the cut-off in order to minimize the subjective perception of the probability scales by radiologists.

## 3. Results

### 3.1. Chest X-ray

The ROC analysis results for all readers and the AI models are shown in Figure 3a. The threshold for AI was 0.23 (Youden index was 0.75); for human readers, the threshold was 3 (Youden index was 0.84). The AI model achieved an AUROC of 0.92 (0.85–0.98), while for radiologists, the AUROC was 0.97 (0.94–1.0). In most regions of the ROC curve, AI performed a little worse than an average human reader or at the same level, but without a statistically significant difference (Table 2). AI accuracy metrics appeared to be the most similar when radiologists’ had less than one year of experience (*p* = 0.76), as shown in Table 2. Examples of the discrepancy between the ground truth and radiologists’ opinions and/or AI results are shown in Figure 4a–c.

### 3.2. Chest Digital Fluorography (FLG)

The ROC analysis results for all readers and the AI models are shown in Figure 3b. The threshold for AI was 0.5 (Youden index was 0.61); for human readers, the threshold was 4 (Youden index was 0.87). A cutoff of 3 (the median of 5-point scale) would reduce the specificity of radiologists to 0.82 (0.79–0.90) but would increase their sensitivity to 1.0 (1.0–1.0). The AI models achieved an AUROC value of 0.83 (0.76–0.9), while radiologists had a higher AUROC of 0.98 (0.96–1.00). The difference between AI and radiologists was statistically significant. Similar to the X-ray use case, AI results were the most comparable to those radiologists with minimal experience (*p* = 0.25) (Table 2). An example of discrepancies in the opinions of AI and radiologists is shown in Figure 4d.

### 3.3. Mammography (MMG)

The ROC analysis results for all readers and the AI model are illustrated in Figure 3c. The threshold for AI was 0.60 (Youden index was 0.65); for human readers, the threshold it was 3 (Youden index was 0.81). The ROC analysis results for a subgroup of readers, breast imaging specialists and the AI model are shown in Figure 3d. The threshold for breast imaging specialist readers was 3 (Youden index was 0.85). The AI models achieved an AUROC of 0.89 (0.83–0.94) that was lower (*p* < 0.05) than the AUROCs of both general radiologists (0.94 (0.91–0.97)) and breast imaging specialists (0.96 (0.93–0.99)). For both radiologist subgroups, AI models were comparable to the radiologists with less than 5 years of experience. It is worth noting that the dataset used in this study contained an atypical ratio of pathological and normal findings, which could affect the breast imaging specialists’ analysis. Therefore, due to their oncological alertness, there was a slight decrease in the average value of specificity. This statement is illustrated by the example in Figure 5.

### 3.4. Overall

Comparing the average results of the eight AI models, as well as an average assessment of radiologists’ performance, the significant differences (*p* < 0.05) in the AUROC values of 0.87 (0.83–0.9) for AI algorithms vs. 0.96 (0.94–0.97) for radiologists were obtained (Figure 3e, Table 2). For the cut-off values calculated by Youden’s method, lower values of sensitivity and specificity (95% CIs did not intersect) were detected for AI than for radiologists. Comparison of radiologists’ performance with AI revealed the best match with the group of least experienced radiologists (*p* = 0.65) (Table 2). Diagnostic accuracy metrics per AI model are shown in the Appendix A.

## 4. Discussion

The diagnostic accuracy of any method, including an AI model, is a key parameter when making a decision regarding the practical applicability of this method in medicine. Some studies demonstrated an increase in the diagnostic accuracy metrics of a radiologist using an AI model [4,34]. Our work aimed to benchmark radiologists and AI in interpreting images independently. Currently, the question concerning the applicability of AI to interpret screening studies remains open. Several studies have demonstrated the high diagnostic accuracy of AI algorithms for screening studies such as mammography [13,30,31,32] and chest X-ray [7,18,33]. In a double reading setting, such as in Europe, highly accurate AI models could alleviate the person-power needed for the interpretation, reaching the consensus of two radiologists. However, several studies [4,17,34,40] have indicated a lack of accuracy of AI algorithms. In this study, we compared the average accuracy of AI screening models with the average accuracy of radiologists. This study is important because it not only evaluated the diagnostic accuracy of radiologists in various types of examinations, but also compared the performance of general radiologists with that of mammography specialists.

A collective performance assessment was applied to the groups of radiologists with different levels of experience to estimate the necessary requirements for AI to be able to substitute human readers for mass screening and routine examinations. The collective evaluation of radiologists’ responses demonstrated the variability of their accuracy depending on their experience and specialization. A similar calculation was also carried out for all participating AI models to calibrate and combine their responses. The use of the binary criteria ‘with pathology’ and ’without pathology‘ allowed the unification of the assessment criteria for radiologists and AI, and ensured their objective comparison for the triage and detection tasks (identifying pathological changes in a study); however, it did not facilitate the comparison of accuracy in solving a classification and differential diagnosis task. It is currently strongly recommended to compare the AI models and healthcare professionals on the same datasets for an objective comparison [41]. In our study, all results were received on the same samples. The values of the diagnostic accuracy metrics obtained in this study could be used as a threshold for a successful validation of adaptive AI models during the acceptance tests [42].

A comparative accuracy assessment of the detection of pathological signs in mammography and fluorographic images between radiologists and AI showed that the diagnostic accuracy metrics of radiologists exceeded those of AI, and this was statistically significant. However, for chest X-ray, our study showed no statistically significant differences between AI and radiologists, which was consistent with the results of Wu et al. [18]. This implies the potential for using AI algorithms for the preliminary interpretation of chest X-ray under conditions of a staff shortage [18]. Regarding mammography, a study by McKinney et al. [14] was indicative, in which the AI model was not inferior in performance to breast imaging specialists and allowed radiologists to reduce the workload on doctors by up to 88%. The other group [4] came to similar conclusions. In contrast to the results of the work of McKinney S.M. [14], in our study, the accuracy metrics of the average radiologists were higher than those of the AI models. None of the AI models surpassed radiologists in this study. However, this study emphasizes the potential of using machine learning techniques to improve the interpretation of screening mammography by radiologists without significant experience in breast imaging.

In the present study, the data obtained clearly demonstrated that AI models exhibited superior diagnostic accuracy compared to novice radiologists. This finding aligns with previous research studies that have also reported the effectiveness of AI models in improving diagnostic accuracy [43] The results suggest that AI models have the potential to serve as decision support systems (DSS) for novice radiologists, assisting them in their training and enhancing the quality of their work. In conclusion, the data from this study support the notion that AI models outperform novice radiologists in terms of diagnostic accuracy, which is consistent with previous research. The potential use of AI models as DSS tools for beginners holds promise in improving their training and improving the quality of their work. More research and implementation efforts are needed to explore the optimal integration of AI models into radiology practice and to assess their long-term impact on patient outcomes.

In this study, we conducted a comparison of diagnostic accuracy between a radiologist and AI algorithms. Our findings clearly demonstrated that, as the radiologist’s experience increased, quality indicators also improved. This indicates that with an increasing number of examined studies, the accuracy of the radiologists’ work tends to increase. Similarly, the same observation can be made for AI algorithms: the larger the dataset used for training, the higher the quality of the AI model. To provide context, some researchers have used a significant dataset of 108,948 [44] studies when developing AI for chest radiography. On average, a radiologist interprets 50 examinations per shift. Therefore, over a period of 10 years, a radiologist would review slightly more than 120,000 studies. Consequently, the number of studies evaluated by a radiologist in a 10-year period can be considered comparable to the dataset on which an AI algorithm could be trained.

Due to the lack of radiologists and a continuous increase in the amount of routine radiology examinations, the use of AI will make it possible to revise the healthcare development pathway for radiology [45]. For example, instead of the increase in the number of radiology residents to fulfill primary healthcare needs, one could shift the pathway to the direction of narrower specialization and expertise for the radiologists, while leaving routine screening studies to AI.

## 5. Conclusions

Benchmarking of AI models and radiologists on a multicenter and multinational level demonstrated that the overall accuracy of AI was lower than the accuracy of the radiologists. The AI and radiologists’ performance levels were the most comparable for chest X-ray. In contrast, AI was inferior to human readers for fluorography and mammography. Similar to previous studies, the diagnostic accuracy of AI can be compared with physicians undergoing residency training in radiology. In summary, this study showed that the application of existing AI models for routine and mass screening in radiology is possible as a substitute for residency trainees in their first reading to reduce the workload of radiologists. The diagnostic accuracy metrics for screening methods of the average radiologist obtained in this work can be used as target values in the development, training and fine-tuning of AI algorithms.

## 6. Limitation

The current study was limited by the relatively small size of the dataset; thus, the diagnostic accuracy metrics of separate pathological findings with sufficient statistical significance could not be calculated.

The limitations of the study from the radiologists’ point of view included the lack of clinical information, the limited functionality of the DICOM web viewer, a sample of studies enriched with pathologies that differed from routine practice and a strictly algorithmic probability model for interpreting the studies.

In this work, AI models were used in versions that were relevant at the end of 2021.

## Figures and Tables

**Figure 1 healthcare-11-01684-f001:**
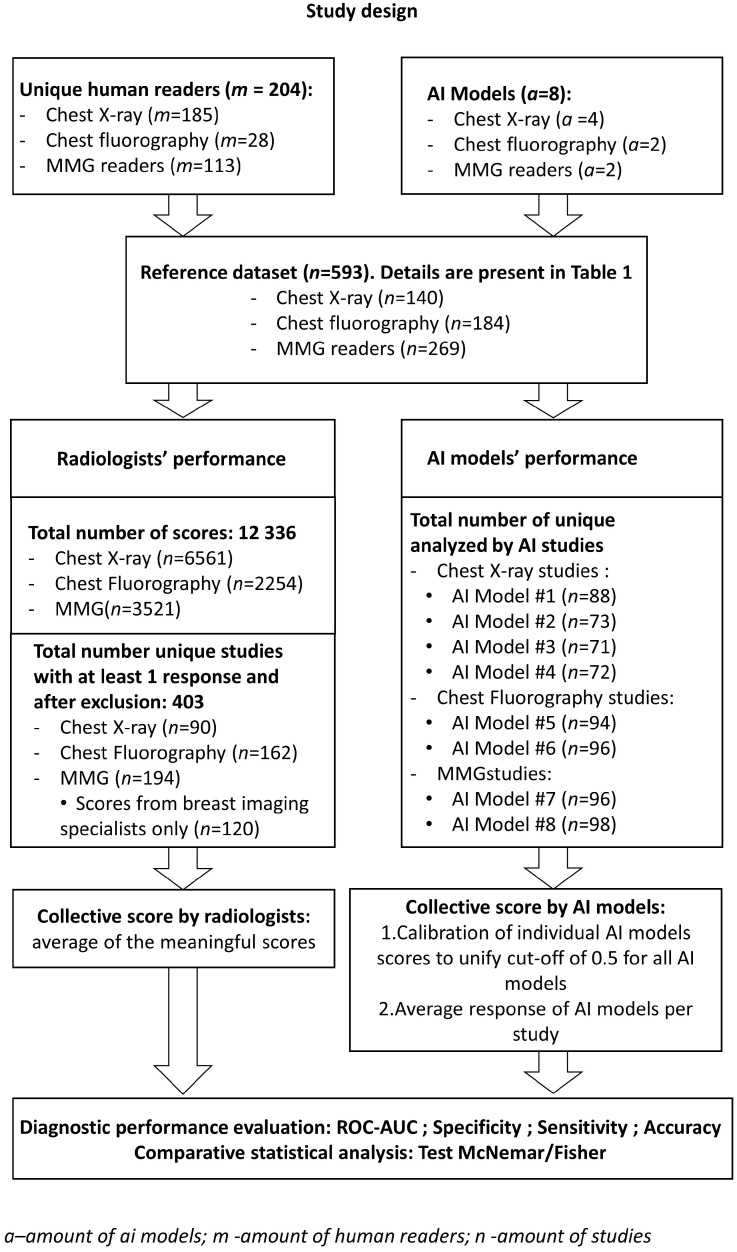
The study design scheme.

**Figure 2 healthcare-11-01684-f002:**
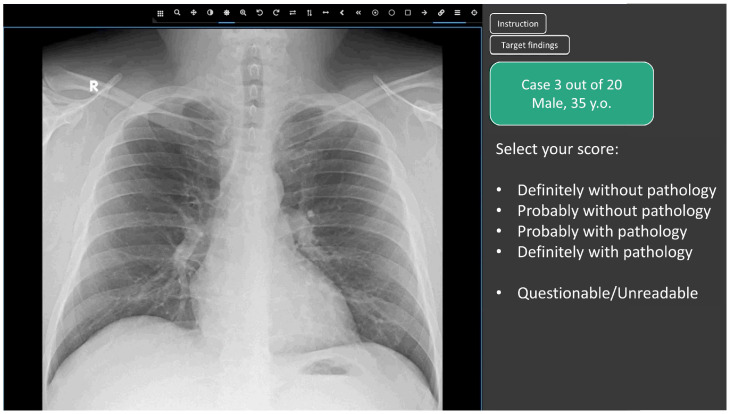
Radiology examination view window of the developed Web platform for the reader study.

**Figure 3 healthcare-11-01684-f003:**
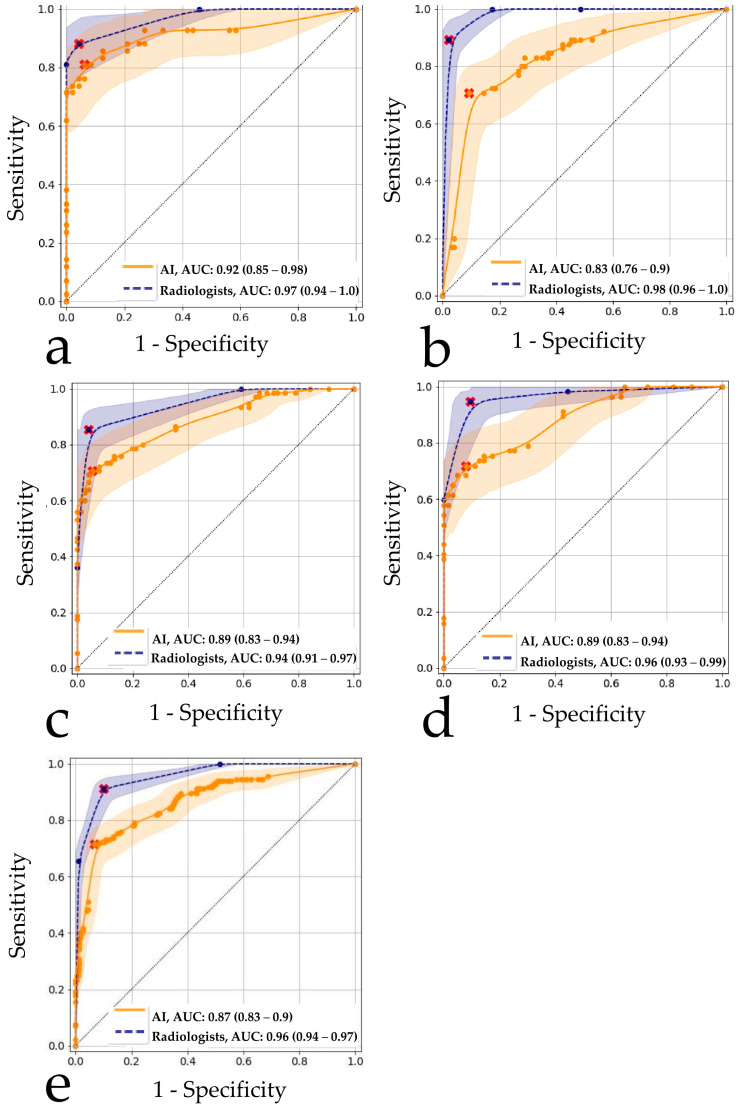
Receiver operator characteristic curves for the human reader study and AI performance on the same datasets: (**a**) Chest X-ray, (**b**) Chest digital fluorography, (**c**) Mammography by all readers, (**d**) Mammography by breast imaging specialists, (**e**) Combined result. Smoothing was used to build the ROC curves. Red markers indicate an operating point determined as the maximum of the Youden index for the readers and AI. The legends display AUC values (95% CI).

**Figure 4 healthcare-11-01684-f004:**
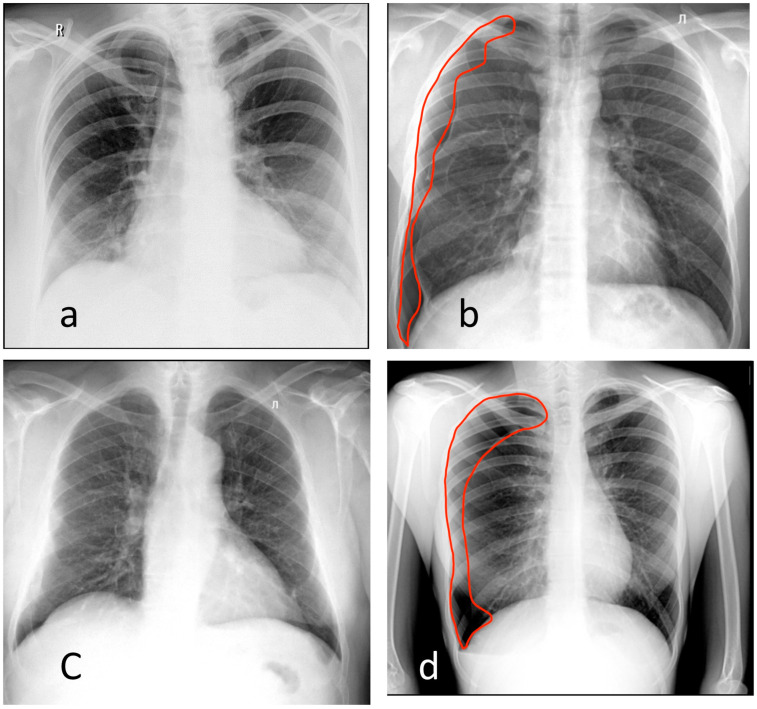
Examples of chest X-rays (**a**–**c**) and digital fluorography (**d**). (**a**) Radiologists misjudged this case as pathological. Increased opacity in the right- and left-sided lower lobe can be mistakenly interpreted as pneumonia without taking into consideration the patient’s suboptimal positioning. The AI models did not detect pathological changes and correctly marked this case as ‘without target pathology‘. (**b**) AI missed a right-sided pneumothorax (red markup). Radiologists correctly marked this case as pathological. (**c**) Both radiologists and AI misjudged this case as pathological. The only confusing findings included calcified lymph nodes in the right hilum and superposition of anatomical structures. (**d**) AI missed a right-sided pneumothorax (red markup). Radiologists correctly marked this case as pathological.

**Figure 5 healthcare-11-01684-f005:**
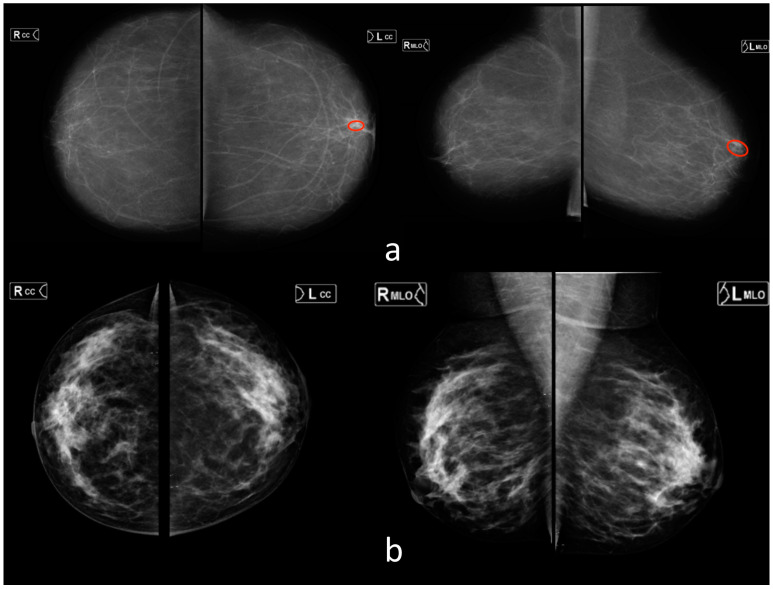
Examples for mammography (**a**,**b**). (**a**) AI missed a cluster of retroareolar microcalcifications in the left breast (red markup). Radiologists correctly marked this case as pathological. (**b**) Areas of disseminated breast fibroglandular tissue (type C according to ACR) were mistakenly interpreted by breast imaging specialists as suspicious for malignancy. AI did not detect pathological changes.

**Table 1 healthcare-11-01684-t001:** Details on the datasets and human readers.

Parameter	X-ray	Fluorography	MMG
Number of cases (cases “with pathology”) *	140 (47)	184 (84)	269 (167)
Confirmation of (ab)normality by	Two experts (>5 years of experience)		
Male/female/unknown	59/77/4	94/113/4	0/269/0
Age (years) **	49 ± 18 [15, 89]	53 ± 19 [19, 93]	63 ± 6 [34, 80]
Radiological findings	Pleural effusion (9) Pneumothorax (7) Atelectasis (9) Nodules or mass (21) Infiltrate or consolidation (13) Miliary pattern, or dissemination (1) Cavity (1) Pulmonary calcification (7) Fracture (2)	1. Pleural effusion (26) 2. Pneumothorax (7) 3. Nodules or mass (28) 4. Infiltrate or consolidation (26) Pulmonary calcification (14)	BiRADS 0
Number of diagnostic devices	61	69	11
Vendors	(1) GE Medical Systems, LLC (2) Fujifilm (3) Toshiba Medical Systems, Inc (4) RENinMED, LLC	(1) S.P. Gelpik, LLC	(1) Fujifilm
Radiologists (number)	185	28	113 (96 breast imaging specialists)
Years of experience			
0–1	36	6	16 (15)
1–5	60	8	32 (28)
5–10	36	5	28 (24)
10+	53	9	37 (29)
Country ***	AM—1 AZ—1 BY—11 GE—1 KG—2 KZ—6 LV—1 MD—2 RU—141 UA—17 UZ—2	BY—1 GE—1 KZ—1 RU—25	AZ—1 BY—4 GE—1 KG—1 KZ—4 LV—1 MD—1 RU—95 UA—4 UZ—1

* Cases “with pathology” contained at least one of the radiological findings. ** Data are mean ± standard deviation. Data in parentheses are the range. *** Codes for countries from ISO 3166.

**Table 2 healthcare-11-01684-t002:** Human and AI diagnostic performance metrics.

Modality		Diagnostic Performance Metrics				
		AUROC (CI 95%)	Sensitivity * (CI 95%)	Specificity * (CI 95%)	Accuracy * (CI 95%)	*p*-Value (for the AUROC)
X-ray	AI n = 90	0.92 (0.85–0.98)	0.81 (0.66–0.91)	0.94 (0.83–0.99)	0.88 (0.79–0.94)	-
	Radiologists (all) n = 90	0.97 (0.94–1.0)	0.88 (0.74–0.96)	0.96 (0.86–0.99)	0.92 (0.85–0.97)	0.104
	Radiologists (0–1 year) n = 83	0.87 (0.79–0.95)	0.74 (0.57–0.87)	0.96 (0.85–0.99)	0.86 (0.76–0.92)	0.76
	Radiologists (1–5 years) n = 86	0.92 (0.86–0.99)	0.83 (0.69–0.93)	1.00 (0.92–1.00)	0.92 (0.84–0.97)	0.43
	Radiologists (5–10 years) n = 65	0.93 (0.86–1.00)	0.83 (0.65–0.94)	0.97 (0.85–1.00)	0.91 (0.81–0.97)	0.40
	Radiologists (10+ years) n = 84	0.98 (0.96–1.00)	0.92 (0.79–0.98)	0.93 (0.82–0.99)	0.93 (0.85–0.97)	0.08
FLG	AI n = 162	0.83 (0.76–0.9)	0.71 (0.58–0.81)	0.91 (0.83–0.96)	0.83 (0.76–0.88)	-
	Radiologists (all) n = 162	0.98 (0.96–1.00)	0.89 (0.79–0.96)	0.98 (0.93–1.00)	0.94 (0.90–0.97)	0.00 **
	Radiologists (0–1 year) n = 14	0.96 (0.87–1.00)	1.00 (0.63–1.00)	0.83 (0.36–1.00)	0.93 (0.66–1.00)	0.25
	Radiologists (1–5 years) n = 42	0.99 (0.98–1.00)	0.91 (0.72–0.99)	1.00 (0.82–1.00)	0.95 (0.84–0.99)	0.06
	Radiologists (5–10 years) n = 12	1.00 (1.00–1.00)	1.00 (0.48–1.00)	1.00 (0.59–1.00)	1.00 (0.74–1.00)	0.09
	Radiologists (10+ years) n = 27	0.97 (0.90–1.00)	1.00 (0.74–1.00)	0.93 (0.68–1.00)	0.96 (0.81–1.00)	0.04 **
MMG General Radiologists	AI n = 151	0.89 (0.83–0.94)	0.71 (0.59–0.81)	0.95 (0.87–0.99)	0.83 (0.76–0.88)	-
	Radiologists (all) n = 151	0.94 (0.91–0.97)	0.85 (0.75–0.92)	0.96 (0.89–0.99)	0.91 (0.85–0.95)	0.01 **
	Radiologists (0–1 year) n = 15	0.91 (0.77–1.00)	0.88 (0.47–1.00)	0.86 (0.42–1.00)	0.87 (0.60–0.98)	0.33
	Radiologists (1–5 years) n = 62	0.92 (0.87–0.98)	0.77 (0.56–0.91)	0.94 (0.81–0.99)	0.87 (0.76–0.94)	0.15
	Radiologists (5–10 years) n = 35	0.90 (0.78–1.00)	0.83 (0.59–0.96)	0.94 (0.71–1.00)	0.89 (0.73–0.97)	0.26
	Radiologists (10+ years) n = 55	0.97 (0.93–1.00)	0.91 (0.72–0.99)	1.00 (0.89–1.00)	0.96 (0.87–1.00)	0.02 **
MMG Breast Imaging Radiologists	AI n = 120	0.89 (0.83–0.94)	0.72 (0.58–0.83)	0.92 (0.82–0.97)	0.82 (0.75–0.89)	-
	Breast Imaging Radiologists (all) n = 120	0.96 (0.93–0.99)	0.95 (0.85–0.99)	0.90 (0.80–0.96)	0.93 (0.86–0.97)	0.01 **
	Breast Imaging Radiologists (0–1 year) n = 13	0.85 (0.66–1.00)	1.00 (0.63–1.00)	0.60 (0.15–0.95)	0.85 (0.55–0.98)	0.52
	Breast Imaging Radiologists (1–5 years) n = 36	0.88 (0.76–1.00)	0.93 (0.68–1.00)	0.71 (0.48–0.89)	0.81 (0.64–0.92)	0.55
	Breast Imaging Radiologists (5–10 years) n = 58	0.92 (0.85–0.98)	0.88 (0.68–0.97)	0.74 (0.56–0.87)	0.79 (0.67–0.89)	0.10
	Breast Imaging Radiologists (10+ years) n = 109	0.97 (0.94–1.00)	0.91 (0.80–0.97)	0.96 (0.87–1.00)	0.94 (0.87–0.97)	0.001 **
Overall result (X-ray + FLG + MMG)	AI solutions (All) n = 403	0.87 (0.83–0.9)	0.71 (0.64–0.78)	0.93 (0.89–0.96)	0.83 (0.79–0.87)	
	Radiologists and Breast Imaging Radiologists (All) n = 403	0.96 (0.94–0.97)	0.91 (0.86–0.95)	0.90 (0.85–0.94)	0.91 (0.87–0.93)	0.00 **
	Radiologists (0–1 year) n = 112	0.93 (0.89–0.97)	0.93 (0.8–0.98)	0.80 (0.69–0.89)	0.85 (0.77–0.91)	0.65
	Radiologists (1–5 years) n = 190	0.94 (0.91–0.97)	0.89 (0.81–0.94)	0.87 (0.79–0.92)	0.87 (0.83–0.91)	0.02 **
	Radiologists (5–10 years) n = 112	0.95 (0.92–0.98)	0.90 (0.81–0.96)	0.90 (0.82–0.95)	0.90 (0.85–0.94)	0.23
	Radiologists (10+ years) n = 166	0.97 (0.95–0.99)	0.92 (0.86–0.96)	0.91 (0.86–0.95)	0.92 (0.88–0.95)	0.00 **

n, Number of cases. * At the operating point of maximum Youden index. ** Statistically significant difference in AUROC values.

## Data Availability

The data presented in this study are available on request from: [https://mosmed.ai/en/datasets/, accessed on 1 June 2023] or from the corresponding author. The datasets are not publicly available due to them still being used to test AI systems participating in the experiment on the use of innovative computer vision technologies for medical image analysis and their subsequent applicability in the healthcare system of Moscow.

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
