# Peer review of "An International Non-Inferiority Study for the Benchmarking of AI for Routine Radiology Cases: Chest X-ray, Fluorography and Mammography"

_healthcare, 2023, doi:10.3390/healthcare11121684_

Round 1

Reviewer 1 Report

The authors performed a comprehensive study to compare the radiologist and AI models' performances. Generally, the paper is well-written. I believe that if the authors provide the details of the following points, it would be one of the reference studies.

-          The main contribution or novelty needs a better explanation in the introduction. The authors referred to the studies among 101 radiologists from 7 countries (for breast cancer with mammography), 1100 participants from 44 countries for screening mammography, and the last one with six readers. Is the contribution the number of radiologists (204) and countries (11)? Additionally, this should be clearly indicated if the study distinguished from others by using both chest X-ray and mammography images.

-          It would be better if the details of AI models were provided in a separate table.

-          The AI experiments are not clear. No details were provided. For example, did the authors train these models or use pre-trained weights for their study? If they trained the models, all details should be included for the reproducibility of the study.

-          The authors mentioned that the classification option was 'without pathology' and 'with pathology'. However, the scores of human readers were obtained with five options on the web platform. How did the authors get the scores of 5 options from AI models if they were able to classify the images for binary targets? It was mentioned that the mean of the probability scores of AI models was used. Were these scores for the five options?

-          Even though they are not limitations, I think some points limit the generability of the study. It is the comparison of AI models and radiologists. I am curious how many images were used to train (experience) the AI models and how many cases the radiologist analyzed in their experience group (i.e., 10+ experienced). Therefore, it could be beneficial to provide the average number of cases and discuss this difference (AI experience and radiologists' experience) in the discussion section.

-          The authors mentioned in the abstract that the AI models could outperform the least experienced radiologists. However, I believe it will be more useful if the authors discuss it in more detail.

-          The radiologists were categorized based on their experience, even though all scores were used to assess the performance. However, the scores of AI models were obtained by considering the mean scores of all models. Providing the top-scored AI model and comparing its' performance separately with human readers might also be useful to determine if some AI models could really outperform human beings or not.

Author Response

We would like to thank the reviewers for their valuable comments regarding the manuscript. We have carefully considered the comments and addressed them with point-by-point responses. All corresponding modifications in the manuscript have been marked.

Response to Reviewer 1 Comments

The authors performed a comprehensive study to compare the radiologist and AI models' performances. Generally, the paper is well-written. I believe that if the authors provide the details of the following points, it would be one of the reference studies.

Point 1.          The main contribution or novelty needs a better explanation in the introduction. The authors referred to the studies among 101 radiologists from 7 countries (for breast cancer with mammography), 1100 participants from 44 countries for screening mammography, and the last one with six readers. Is the contribution the number of radiologists (204) and countries (11)? Additionally, this should be clearly indicated if the study distinguished from others by using both chest X-ray and mammography images.

Response 1: We thank the reviewer for the suggestion. The existing body of research on the diagnostic accuracy of radiologists in the field of radiology has primarily focused on specific imaging modalities, such as mammography or radiography. However, there is a lack of studies that comprehensively evaluate the diagnostic accuracy of the same radiologists across multiple imaging modalities. In addition, mammography, due to its inherent complexity, often requires specialized expertise, leading to the establishment of a separate subspecialty within medical organizations.

This study is important as it not only evaluates the diagnostic accuracy of radiologists in various types of examinations but also compares the performance of general practitioner radiologists with that of specialists in mammography.

Point 2.               It would be better if the details of AI models were provided in a separate table.

Response 2: We thank the reviewer for the suggestion. In this study, we utilized pre-existing commercial models to assess their performance. Due to proprietary reasons, the developers have not fully disclosed detailed information about the architecture of the AI algorithms and the specific datasets used for training. This lack of transparency is primarily due to the protection of trade secrets.

From the available literature sources, we were able to gather limited information about the AI algorithm architecture and training datasets, which we have included in a separate table in in the supplementary material. However, it is important to acknowledge that this extracted information may not be entirely reliable or up to date. The developers are continually refining their algorithms, and it is possible that the AI models used in our study may have undergone modifications in their architecture or received additional training.

Therefore, while we have made efforts to provide relevant information about the AI models used, it is crucial to recognize the potential limitations and uncertainties associated with the disclosed details. The proprietary nature of these commercial models restricts the full disclosure of their inner workings, and our understanding is based on the available literature at the time of this study.

Point 3.          The AI experiments are not clear. No details were provided. For example, did the authors train these models or use pre-trained weights for their study? If they trained the models, all details should be included for the reproducibility of the study.

Response 3: We thank the reviewer for the suggestion. In the present study, we did not conduct any refinement of the AI models. Instead, we exclusively utilized off-the-shelf commercial solutions as they were provided by the developers. It is important to note that no modifications or alterations were made to the AI models during the course of this study.

However, it is worth mentioning that for constructing the characteristic curves, we employed the probability values of the presence of pathology obtained from the study, rather than relying solely on the output of the classifier. This approach allowed us to gather a more comprehensive understanding of the diagnostic performance across different thresholds

Point 4.          The authors mentioned that the classification option was 'without pathology' and 'with pathology'. However, the scores of human readers were obtained with five options on the web platform. How did the authors get the scores of 5 options from AI models if they were able to classify the images for binary targets? It was mentioned that the mean of the probability scores of AI models was used. Were these scores for the five options?

Response 4: We thank the reviewer for the comment The primary method used for analysis in this study was ROC (Receiver Operating Characteristic) analysis. To conduct the ROC analysis, we required a binary estimation (true value) as well as the output from the "classifier." When evaluating AI algorithms, we utilized the pathology probability value as input, which ranged from 0 to 1 with a precision of 0.01. Similarly, when evaluating a doctor's performance, we also employed the probability values assigned by the doctor. However, it is challenging for a doctor to precisely assign a digital probability value for the presence of pathology. Therefore, we employed a more comprehensible gradient scale that could be easily converted into absolute values: "definitely without pathology" = 0, and "definitely with pathology" = 1. Subsequently, a standard ROC analysis was conducted to determine the diagnostic accuracy indicators.

Point 5-          Even though they are not limitations, I think some points limit the generability of the study. It is the comparison of AI models and radiologists. I am curious how many images were used to train (experience) the AI models and how many cases the radiologist analyzed in their experience group (i.e., 10+ experienced). Therefore, it could be beneficial to provide the average number of cases and discuss this difference (AI experience and radiologists' experience) in the discussion section.

Response 5: Indeed, the question regarding the number of chest x-rays a radiologist views in 10 years is intriguing. Unfortunately, we do not possess precise data as we did not conduct specific interviews with physicians, and the datasets used to train the AI models remain a trade secret. However, we can make an estimation based on a hypothetical scenario. Let us assume that one of the largest datasets, such as NIH, which comprises 108,948 studies [X. Wang, Y. Peng, L. Lu, Z. Lu, M. Bagheri and R. M. Summers, "ChestX-Ray8: Hospital-Scale Chest X-Ray Database and Benchmarks on Weakly-Supervised Classification and Localization of Common Thorax Diseases," 2017 IEEE Conference on Computer Vision and Pattern Recognition (CVPR), Honolulu, HI, USA, 2017, pp. 3462-3471, doi: 10.1109/CVPR.2017.369..], was used to train the AI model. On average, a radiologist interprets approximately 50 examinations per shift. Consequently, over a 10-year period, a radiologist would interpret slightly more than 120,000 studies. Therefore, the number of studies reviewed by a radiologist over a 10-year period would be comparable to the dataset on which the AI algorithm could have been trained.

Point 6-          The authors mentioned in the abstract that the AI models could outperform the least experienced radiologists. However, I believe it will be more useful if the authors discuss it in more detail.

Response 6: We thank the reviewer for the comment. In the present study, the data obtained clearly demonstrate that AI models exhibited superior diagnostic accuracy compared to novice radiologists. This finding aligns with previous research studies that have also reported the effectiveness of AI models in improving diagnostic accuracy [Dratsch, T., Chen, X., Rezazade Mehrizi, M., Kloeckner, R., Mähringer-Kunz, A., Püsken, M., ... & Pinto dos Santos, D. (2023). Automation Bias in Mammography: The Impact of Artificial Intelligence BI-RADS Suggestions on Reader Performance. Radiology, 222176.]. The results suggest that AI models have the potential to serve as decision support systems (DSS) for novice radiologists, assisting them in their training and enhancing the quality of their work. In conclusion, the data from this study support the notion that AI models outperform novice radiologists in terms of diagnostic accuracy, consistent with previous research. The potential use of AI models as DSS tools for beginners holds promise in improving their training and improving the quality of their work. More research and implementation efforts are needed to explore the optimal integration of AI models into radiology practice and to assess their long-term impact on patient outcomes.

Point 7-          The radiologists were categorized based on their experience, even though all scores were used to assess the performance. However, the scores of AI models were obtained by considering the mean scores of all models. Providing the top-scored AI model and comparing its' performance separately with human readers might also be useful to determine if some AI models could really outperform human beings or not.

Response 7: We thank the reviewer for the comment. We have presented the diagnostic accuracy data for all AI-based algorithms considered in this paper in separate sections (see the supplementary materials). It is worth noting that the results of the best-performing algorithms, both on a limited data set and in real-time scenarios, have been previously described in our earlier work (https://doi.org/10.3390/diagnostics13081430).

Reviewer 2 Report

In the proposed study a comparison is made between Expert Radiologist and Machine learning techniques. The study is comprehensive and most of the details of the setup are explained. There are a few typos and the style of English can be improved. As an example, the space between words and a few references such as reference numbers 22, 24 is missing. In Line 85, the statement seems incomplete and verbs should be used to complete it.

The Figure 1 for study design seems too occupied with texts and it may be divided into sub figures with one figure should show overall comparison between readers and AI and second figure should present a few details only.

How the ground truth is established and various classes are defined.

In AI models, the prominent classifiers used in AI models, for example, SVM, ANN, Random Forest should be mentioned and the one who is superior in performance should be mentioned.   

Apart from ROC curves, please draw confusion matrices for the comparisons.

In Figure 4, the missed judgements should be pointed out with arrows.

Recent AI techniques, especially deep learning models have superior performances but they need large amount of data and images for training. Please mention in the future directions to perform an analysis with deep learning frameworks. 

The level of English grammar is satisfactory.

Author Response

Thank you for the valuable feedback provided by the evaluator(s) on our manuscript titled. We have carefully addressed all the comments and suggestions, making the necessary revisions to improve the manuscript.

Point 1. In the proposed study a comparison is made between Expert Radiologist and Machine learning techniques. The study is comprehensive and most of the details of the setup are explained. There are a few typos and the style of English can be improved. As an example, the space between words and a few references such as reference numbers 22, 24 is missing. In Line 85, the statement seems incomplete and verbs should be used to complete it.

Response1: We apologize for the confusion. We carefully reread the text of the article and corrected the errors.

Point 2 The Figure 1 for study design seems too occupied with texts and it may be divided into sub figures with one figure should show overall comparison between readers and AI and second figure should present a few details only.

Response2: Thank you for your comments! We have taken your advice into consideration and opted to simplify the study design scheme. As a result, we have transferred some of the information to tables, which has significantly reduced the amount of text in the design scheme.

Point 3 How the ground truth is established and various classes are defined.

Response3: Thank you for your comments! This information is in Reference Dataset section. All studies were selected based on electronic medical records and then double-confirmed by radiology experts with at least 5 years of experience in thoracic radiology or breast imaging. Pathomorphological confirmation for malignancies was obtained from electronic medical records. Please refer to Table 1 for further details.

Point 4 In AI models, the prominent classifiers used in AI models, for example, SVM, ANN, Random Forest should be mentioned and the one who is superior in performance should be mentioned.   

Response4: We appreciate the reviewer's acknowledgment of the practical importance of the information regarding the development of AI models. However, it is important to note that in our study we utilized commercially available solutions, and thus, the specific model architectures and training datasets are considered trade secrets by the developers. Despite this limitation, we made efforts to gather additional information from open sources, which we have included in the supplementary section of the manuscript. However, it is important to emphasize that this detailed information may not be significant to the end user, such as a doctor. From their perspective, an AI-based decision is often perceived as a black box, where the primary focus is on the presented result of the study processing.

Point 5. Apart from ROC curves, please draw confusion matrices for the comparisons.

Response5: Thank you for your comments To avoid overloading the main text of the manuscript, we have included this information in the appendix section. However, we believe it is more informative to present the results using terms that are more understandable to doctors, such as sensitivity, specificity, and accuracy, rather than using a matrix of confusion. The paper provides specific values for the number of cases with pathology and normal cases, making it easy to translate these values into a four-field table format to calculate sensitivity and specificity.

Point 6. In Figure 4, the missed judgements should be pointed out with arrows.

Response6: We apologize for the confusion. We made the necessary corrections to the figures by outlining the pathology with a red line.

Point 7 Recent AI techniques, especially deep learning models have superior performances but they need large amount of data and images for training. Please mention in the future directions to perform an analysis with deep learning frameworks. 

Response7: Thank you for your comments! Many developers are currently utilizing various frameworks to construct their AI solutions. In the scope of this study, we employed comprehensive commercially available research solutions. It is important to note that technology is constantly evolving, and updated versions of the AI algorithms described here may already exist. Nevertheless, our findings indicate that different AI algorithms approach clinical tasks in distinct manners. Hence, the future of AI lies in the utilization of ensemble models, where multiple algorithms work collaboratively to enhance performance and decision-making accuracy. By combining the strengths of different models, ensembles can offer improved outcomes and robustness in clinical applications.

Round 2

Reviewer 1 Report

Thanks to the authors for considering my concerns in the revised version. All of my concerns are well-addressed.